# The influence of lightweight wearable resistance on whole body coordination during sprint acceleration among Australian Rules football players

**Karl M. Trounson**[ID]*, **Sam Robertson**, **Kevin Ball**

Institute for Health and Sport, Victoria University, Footscray, Victoria, Australia

☯ These authors contributed equally to this work.
* karl.trounson@live.vu.edu.au

**Data Availability Statement:** All data are held in a public repository located at: https://datadryad.org/stash/dataset/doi:10.5061/dryad.p2ngf1vxf.

## Abstract

Rapid acceleration is an important quality for field-based sport athletes. Technical factors contribute to acceleration and these can be deliberately influenced by coaches through implementation of constraints, which afford particular coordinative states or induce variability generally. Lightweight wearable resistance is an emerging training tool, which can act as a constraint on acceleration. At present, however, the effects on whole body coordination resulting from wearable resistance application are unknown. To better understand these effects, five male Australian Rules football athletes performed a series of 20 m sprints with either relatively light or heavy wearable resistance applied to the anterior or posterior aspects of the thighs or shanks. Whole body coordination during early acceleration was examined across eight wearable resistance conditions and compared with baseline (unresisted) acceleration coordination using group- and individual-level hierarchical cluster analysis. Self-organising maps and a joint-level distance matrix were used to further investigate specific kinematic changes in conditions where coordination differed most from baseline. Across the group, relatively heavy wearable resistance applied to the thighs resulted in the greatest difference to whole body coordination compared with baseline acceleration. On average, heavy posterior thigh wearable resistance led to altered pelvic position and greater hip extension, while heavy anterior thigh wearable resistance led to accentuated movement at the shoulders in the transverse and sagittal planes. These findings offer a useful starting point for coaches seeking to use wearable resistance to promote adoption of greater hip extension or upper body contribution during acceleration. Importantly, individuals varied in how they responded to heavy thigh wearable resistance, which coaches should be mindful of.

## Introduction

The ability of athletes to rapidly accelerate is an important quality required in many field-based sports. On average, soccer, rugby league, and Australian football athletes perform

**Funding:** The authors received no specific funding for this work.

**Competing interests:** The authors have declared that no competing interests exist.

between 50–100 acceleration efforts ($>2.87$ m.s$^{-2}$ [1]) throughout match play [2, 3]. In addition, these efforts are often associated with critical events, such as winning the ball or breaking away from an opponent [4, 5]. A key role of strength and conditioning coaches in these sports is therefore to develop the kinetic and kinematic factors that contribute to acceleration performance, such as horizontal and vertical ground reaction impulse (GRI) and body segment positions [6–8].

Training programs aiming to improve sprint acceleration usually incorporate a combination of exercises targeted towards achieving neuromuscular overload, as well as drills to improve sprint technique [9, 10]. While the effects of various overload interventions on acceleration kinetics, kinematics, and performance among field-based athletes have been extensively researched [11–15], less attention has been paid to training drills aimed at altering sprint technique. This is despite findings that highlight the importance of technical execution in acceleration [16, 17]. For example, faster individuals exhibit a greater magnitude anteroposterior component of resultant ground reaction forces (GRFs) during acceleration than slower individuals [16, 17]. This contributes to more forward oriented GRFs and superior performance, despite negligible differences in the magnitude of GRFs between the two groups. This is achieved through foot touchdown more posterior relative to the centre of mass [16]. Coaches may therefore seek to understand the coordination patterns associated with faster acceleration and aim to train technique in accordance with the expression of such patterns.

Traditional strength and conditioning approaches to sprint technique training have often utilised skill deconstruction to target a particular aspect of coordination [18, 19]. Modern skill acquisition perspectives, however, advocate the use of pedagogical approaches, which consider movement as an emergent property of a complex system [20–23]. Constraints on complex systems direct emergent behaviour by limiting the behavioural trajectories that can be adopted [24, 25]. Constraints therefore represent control parameters, which can be manipulated by coaches to influence specific movement behaviours that may benefit performance, or to induce general variability and encourage exploration of movement [26, 27]. Both the use of constraints to shape movement, and the use of constraints to induce variability have demonstrated effectiveness as far as altering coordination patterns and improving performance in a number of sporting tasks [23, 28–30].

While there are innumerable constraints that can be imposed to influence sprint acceleration movement organisation, lightweight wearable resistance (WR) is an increasingly popular training tool with possible applications for this purpose. Modern iterations of WR involve attachment of small weights to body segments, such as the trunk, arms, thighs, and shanks [31–39]. To date, most WR research in sprinting has sought to examine the effects of WR as a movement specific overload stimulus [31–34, 36, 38, 40, 41]. Decrements in sprint speed or changes in whole body spatiotemporal gait parameters are seen as indicating overload has occured [32–34, 36, 38, 41]. The focus on WR as a neuromuscular overload tool overlooks its potential use as a coaching tool to alter coordination in a complex systems-based pedagogical framework. While some studies have hinted at this application, suggesting that WR could reinforce piston-like mechanics required during acceleration for example [35, 42], only a limited number [43–47] have actually investigated joint-level kinematic changes induced by WR. Besides affording favourable movement patterns, it is also conceivable that WR implementation may destabilise preferred patters, inducing movement variability. Variability in training can facilitate adaptability, i.e. task execution across more varied contexts, which is advantageous for field-based athletes encountering dynamic and unpredictable scenarios in match play [48, 49]. Given the lack of predictability about movement alterations in response to WR, the present study adopted a broad analytical approach, which considered changes to continuous time series angle data across multiple joints and planes during early acceleration in

response to WR. Initial characterisation of the whole body coordination changes that occur may offer a starting point for coaches in applied settings interested in using WR in a skill acquisition context. This study therefore aimed to determine the extent and manner of whole body coordination changes during sprint acceleration arising from different WR loading configurations and magnitudes among Australian Rules football players, with consideration for both group-level and within-individual changes.

## Materials and methods

### Participants

Five semi-professional male Australian Rules football players (mean ± SD; age: 21.2 ± 4.1 years; height: 180.6 ± 6.5 cm; body mass: 72.0 ± 4.3 kg) were recruited between July 30 and October 15 2019 for participation in this study. Participants are hereafter denoted as P1-P5. For inclusion in the study, participants were required to be currently playing at a semi-professional level, undertaking structured team training twice per week and match play once per week, and have had no prior experience with WR. On average, players of this level are exposed to 25–40 km total running distance across a week, with 3–5 km of this volume occurring at speeds of 20 km/h or greater [50, 51]. All participants provided written informed consent and were free from musculoskeletal injury at the time of, and in the 6 months prior to, testing. All procedures used in this study complied with the criteria of the declaration of Helsinki and ethical approval was granted by the Victoria University Human Research Ethics Committee (HRE19-020).

### Procedure

**Study design.** Testing was undertaken in ambient temperature (24 ± 2˚C) in the Biomechanics Laboratory at Victoria University, Footscray Park, Melbourne, Australia. Participants attended the laboratory on 10 occasions in total, comprising one familiarisation session, one baseline testing session, and eight WR testing sessions. Each testing session was conducted at the same time (09:00 AM) to minimise the influence of circadian variation and was undertaken at least 48 hours post-previous match play or structured team training. Each testing session was separated by at least 1 week. During testing sessions, participants undertook a warm-up, which consisted of a series of dynamic mobility drills, including; forward lunges with arm reaches, leg swings, lateral lunges, and tiptoe walks, executed as previously described [52]. These drills were followed by four sub-maximal 20 m sprints. The 15-grade Borg rating of perceived exertion (RPE) scale [53] was explained to participants, and instruction was given to perform the four warm-up sprints corresponding to "fairly light", "somewhat hard", "hard", and "very hard" levels of exertion, respectively. Following this, participants performed four maximal 20 m sprints commencing from a stationary position, interspersed with 3 min rest periods. During WR testing sessions, participants were exposed to one of eight unique WR loading configurations and magnitudes when performing sprints 2–4. The order of exposure to each WR loading configuration and magnitude was randomised. In each sprint, 10 m split and 20 m sprint times were recorded and whole body spatiotemporal measures and joint kinematics were captured at the 4 m mark to examine coordination during the early acceleration phase of sprinting [54].

**Experimental setup.** A 20 m section of the Biomechanics Laboratory with Mondo track surface defined the sprint area. Infrared timing gates (Smart Speed, Fusion Sport, Brisbane, Australia) were situated at the 0, 10, and 20 m marks along the sprint area. For each sprint, participants adopted a self-selected 2-point upright starting stance with the front foot 0.9 m behind the starting line. Timing began when the timing gates at the 0 m mark were triggered

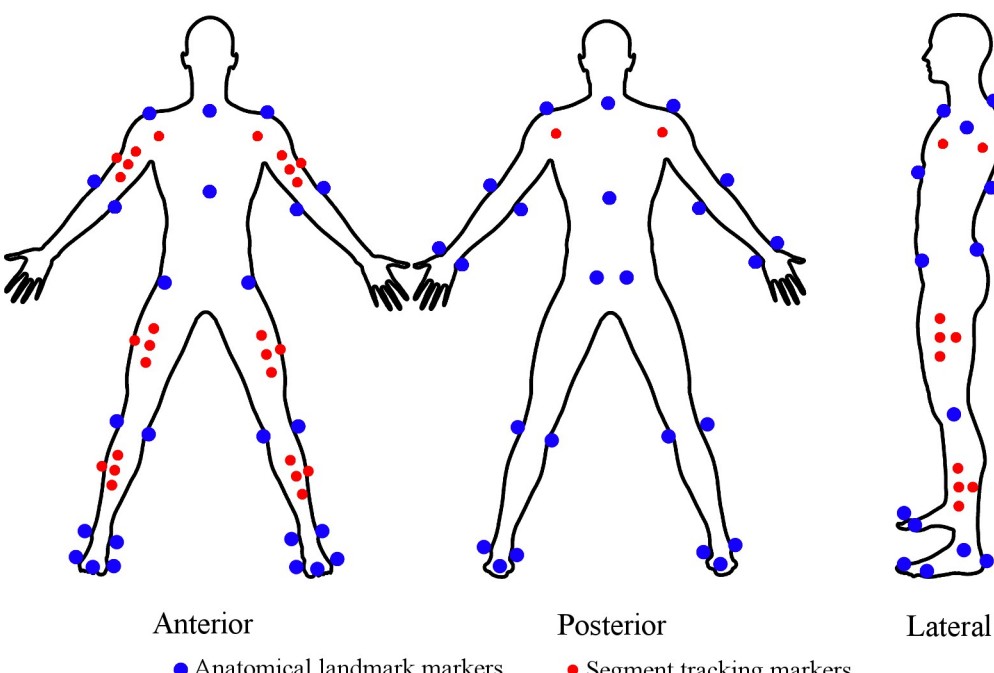

**Fig 1. Upper- and lower-body Plug-In-Gait model marker placements.** Blue markers define the required anatomical landmarks, red markers are used for tracking segments. Adapted from Trounson KM, Busch A, French Collier N, Robertson S (2020) Effects of acute wearable resistance loading on overground running lower body kinematics. PLoS ONE 15(12): e0244361 under a CC BY license, with permission from PLoS ONE, original copyright 2020.

by the participant commencing their sprint. Motion analysis cameras were arranged around the 4 m mark and the approximate capture volume was 5.0 m long, 2.5 m high, and 3.0 m, wide. A 10-camera VICON motion analysis system (T-40 series, Vicon Nexus v2, Oxford, UK) sampling at 250 Hz was used for collection of whole body spatiotemporal and joint kinematic data. A total of 58 reflective markers with 12.7 mm diameter were attached to body landmarks on the upper arms, trunk, pelvis, thighs, shanks, and feet according to the Plug-In-Gait model (Plug-In-Gait Marker Set, Vicon, Oxford, UK) (Fig 1). In Vicon Nexus software, a global reference system was defined with the positive Y-axis horizontal in the direction of the sprint, the positive X-axis perpendicular to the Y-axis–horizontal in the right direction, and the positive Z-axis in the vertical direction.

**Wearable resistance.** Throughout testing, participants wore Lila™ Exogen™ (Sportboleh Sdh Bhd, Kuala Lumpur, Malaysia) compression shorts and calf sleeves. During WR exposure trials, a combination of 50, 100, and 200 g fusiform shaped loads (with Velcro backing) totalling the required loading magnitude were attached to the compression garments (Fig 2). Four loading configurations were investigated–anterior thigh, posterior thigh, anterior shank, and posterior shank–with both "light" and "heavy" loading magnitudes in each, totalling eight WR conditions. "Light" and "heavy" loading magnitudes corresponded to an increase of 3% and 6% in the moment of inertia about the hip throughout an acceleration stride, respectively, in accordance with sagittal plane lower limb motion previously observed during early acceleration [55]. Participant height and weight was used to determine the specific loading magnitudes required at each segment to satisfy these conditions based on Plagenhoef's [56] estimations of segment parameters. Table 1 provides an example of the loading magnitudes per leg for a 180 cm, 70 kg male [56]. Fusiform loads were added at the midpoint of each segment in a longitudinal formation and in an alternating fashion between a proximal-dominant and distal-

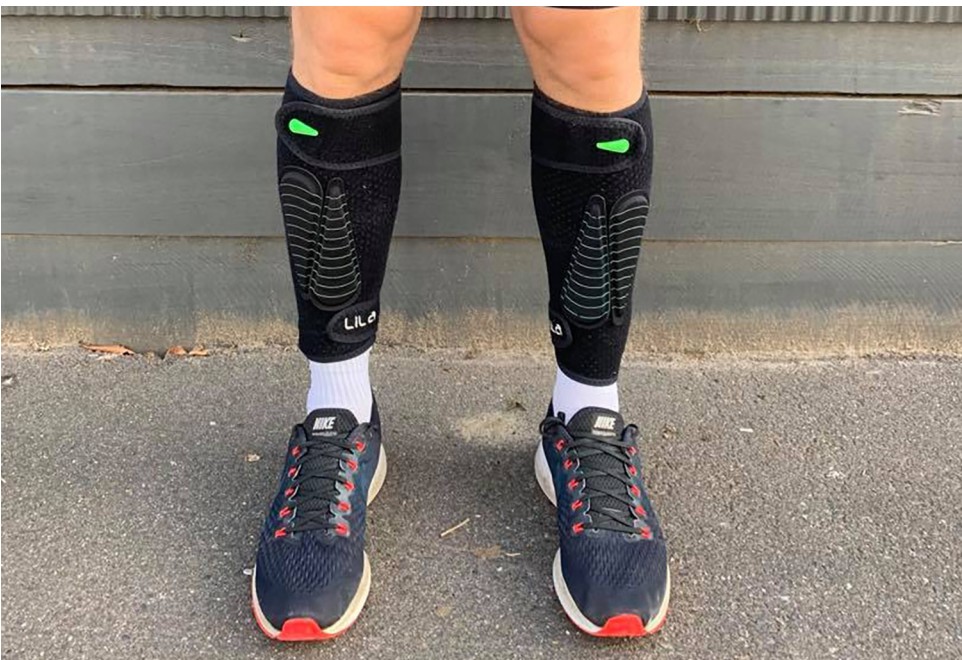

**Fig 2. Lila™ Exogen™ compression calf sleeves with anterior shank loading.** Reprinted from Trounson KM, Busch A, French Collier N, Robertson S (2020) Effects of acute wearable resistance loading on overground running lower body kinematics. PLoS ONE 15(12): e0244361 under a CC BY license, with permission from PLoS ONE, original copyright 2020.

dominant orientation. The smallest number of possible loads to achieve the required loading magnitude was used.

**Data collection.** Following application of compression garments, attachment of reflective markers, and the warm-up, participants performed four maximal 20 m sprints, each separated by 3 mins rest. The only instruction provided to participants was to sprint as fast as possible. In WR testing sessions, researchers applied the requisite WR loads to the participant during the rest period between the first and second sprint, and the WR was left on for the remaining three sprints. Fig 3 provides a summary schematic of the between- and within-testing session structure for a 180 cm, 70 kg participant.

## Data processing

Raw marker data were labelled in Vicon Nexus with cubic spline filling used in instances of marker drop out (up to a maximum of 10 frames). Marker data were then transferred to Visual 3D software (C-motion, Rockville, MD, USA) for calculation of whole body spatiotemporal measures and joint kinematics using the following steps. Marker trajectories were smoothed

**Table 1. Example loading magnitudes for a 180 cm, 70 kg male participant.**

| Configuration | Magnitude | |
|---|---|---|
| | Light (g per leg) | Heavy (g per leg) |
| Anterior thigh | 550 | 1100 |
| Posterior thigh | 550 | 1100 |
| Anterior shank | 250 | 500 |
| Posterior shank | 250 | 500 |

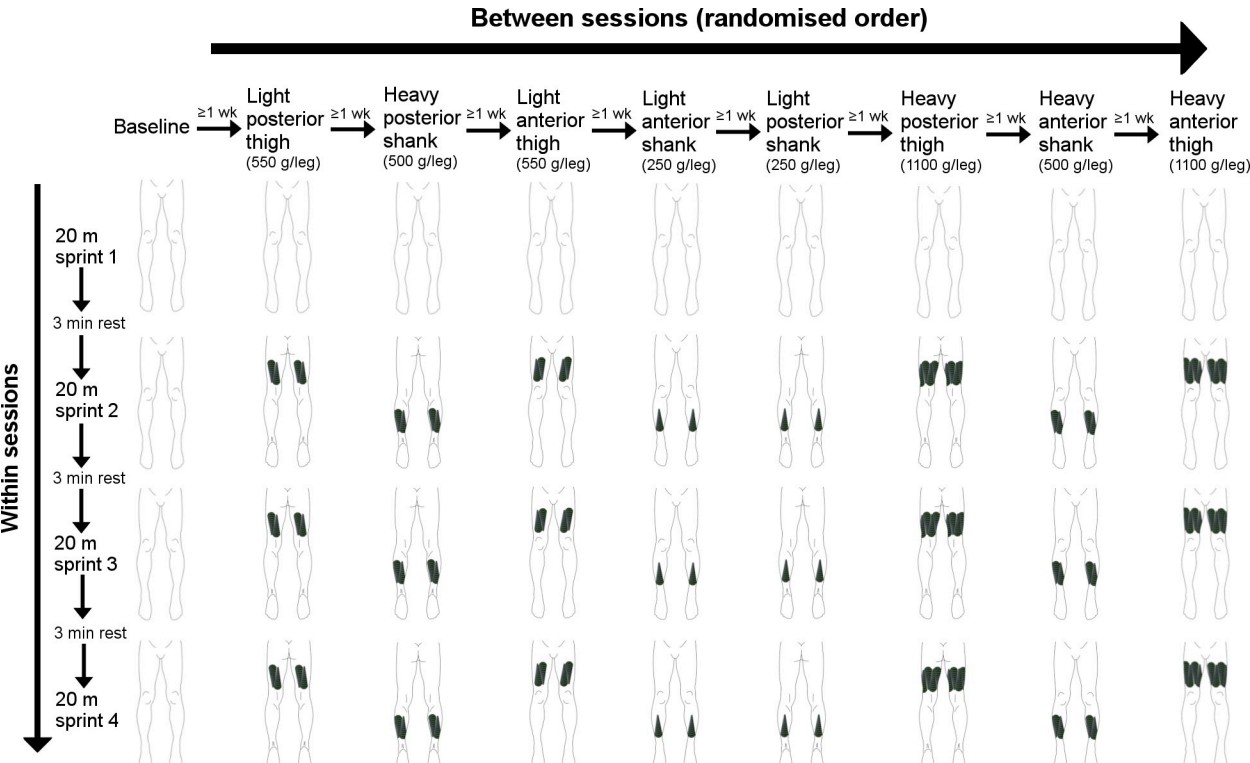

**Fig 3. Summary of the between- and within-testing session structure for a 180 cm, 70 kg participant.** WR conditions were randomised across eight testing sessions separated by at least 1 week. Within testing sessions, participants performed four maximal 20 m sprints interspersed with 3 min rest periods. In all testing sessions the first sprint was performed without WR.

via a fourth order low-pass Butterworth filter with 10 Hz cut-off frequency, based on mean results of residual analyses [57]. A 10-segment model (upper arms, trunk, pelvis, thighs, shanks, and feet) was then constructed for each participant. Each sprint was trimmed to one complete stride cycle, which was defined as the period between two consecutive toe-off events on the same limb. Toe-off was defined by the initial rise in vertical displacement of the toe marker proceeding its lowest point at the end of the stance phase and these timepoints were automatically detected using an event detection algorithm in Visual 3D [58, 59]. Without explicit instruction, all participants chose to commence sprints with the left foot forward. Analysis was therefore able to be carried out on the stride defined by left foot toe-off to left foot toe-off corresponding to steps 3 and 4 of the sprint effort. This stride was taken as representative of the first phase of acceleration identified by Nagahara et al., (2014), who reported, on average, a definitive breakpoint in acceleration kinematics beyond step 4 [60]. Of the 180 captured sprints, only three were unable to be successfully reconstructed according to the above process and these were excluded from analysis. In all instances, sprints 2–4 from each testing session were used when comparing effects between conditions, unless otherwise stated.

For whole body spatiotemporal measures, an in-built model based function in Visual 3D was used to calculate mean centre of mass (COM) velocity across the stride [7]. Flight time was defined as the point of take-off from one foot to the point of ground contact on the contralateral foot. Ground contact time was defined as the point of initial ground contact until the point of take-off on the same foot. Step length was defined as the horizontal distance between successive toe-off events of each contralateral foot. Flight time, ground contact time, and step length were all calculated as an average across the two steps composing the stride cycle [61].

Step frequency was defined as the number of steps taken per second and was calculated as the inverse of stride duration multiplied by two.

For joint kinematics, sagittal, frontal, and transverse plane angles were computed from the transformation between two adjacent segments' local coordinate systems described by an XYZ Cardan sequence of rotations [62]. These computations were performed in Visual 3D using the in-built joint angle model based function. The following joints/segments were included: pelvis, thorax, right and left side hips, knees, ankles, and shoulders. In all cases, proximal segments were used as reference segments, except for the pelvis in which angles were defined in relation to the global reference frame. A total of 30 kinematic variables therefore contributed to defining whole body coordination profiles. All angles were normalised to 101 data points (0–100% of the stride cycle) prior to further analysis.

## Data analysis

**Descriptive statistics.**  Mean and range of 10 m split time, 20 m sprint time, COM velocity at 4 m, and spatiotemporal measures (flight time, ground contact time, step length, and step frequency) were calculated for each participant within conditions and across the five participants.

**Hierarchical agglomerative clustering.**  Hierarchical agglomerative clustering was used to visualise the degree of (dis)similarity of whole body coordination between conditions both at the group level and within-participants. This process yields a dendrogram in which the height of the merger between clusters indicates the degree of similarity or dissimilarity between objects. Objects in this instance were the aggregated kinematic variables collected during sprints in each condition. A higher merger between the aggregated kinematic variables across sprints in two different conditions was considered indicative of greater dissimilarity in whole body coordination between the conditions.

For hierarchical agglomerative clustering of kinematic variables between conditions at the group level, the 30-dimensional vectors (10 joints/segments x 3 degrees of freedom) obtained from joint angles in each sprint were averaged across sprints and participants within each condition (15 sprints per condition in total) to produce nine 30-dimensional input variables (v):

$$v_i = \psi_i, 1(t) \dots, \psi_i, 30(t) \tag{1}$$

where $\psi$k represents the kinematic variable (k = 1, . . ., 30). The index, i, represents the condition (i = 1, . . ., 9), while the time index, t, runs from 0 to 101. Using R (version 3.6.0), data were scaled and a distance matrix was created using the Euclidean distance dissimilarity measure. The single linkage hierarchical clustering algorithm was used to generate the clustering hierarchy and dendrogram. Code for this analysis is provided at https://github.com/ktrounson/WR-acceleration/blob/main/Group-hclust.

Hierarchical agglomerative clustering of kinematic variables between conditions within-individuals followed the same process as above, however, vectors were averaged only across sprints in each condition (heavy anterior thigh, light anterior thigh, heavy posterior thigh, light posterior thigh, heavy anterior shank, light anterior shank, heavy posterior shank, light posterior shank). Averaging three sprints per condition within each participant produced 45 30-dimensional input variables. Code used for this analysis is provided at https://github.com/ktrounson/WR-acceleration/blob/main/Indiv-hclust.

**Self-organising map.**  A self-organising map (SOM) analysis was used to investigate whole body coordination profiles across the course of the stride cycle [63]. The SOM effectively represents the whole body coordination throughout the stride cycle for each participant on a two-dimensional grid. Patterns that are similar to one another in the original kinematic

space are mapped closer to one another in the two-dimensional SOM space. Following scaling, each sprint was inputted as a 30-dimensional vector into the SOM algorithm available in the R *kohonen* package [64]. The training process adopted a linearly decreasing learning rate from α = 0.05 to α = 0.01 and a Gaussian neighbourhood function. The final SOM was projected on a 40x40 hexagonal lattice output space and visualised as a unified distance matrix (U-matrix). The SOM code used is provided at https://github.com/ktrounson/WR-acceleration/blob/main/SOM. In the U-matrix, cells are shaded based on the distances to immediate neighbours. Darker shaded areas have a smaller distance to neighbours and correspond with greater convergence of movement patterns in these areas. The two WR conditions in which there was the most dissimilar whole body coordination compared with baseline (based on the results of hierarchical clustering) were considered to be of particular interest for further analysis and discussion. Best-matching unit trajectories for each participant in these conditions were included in the results section, while best-matching unit trajectories for each participant in the remaining conditions were included as supplementary figures.

**Joint-level distance matrix.** For the two most dissimilar conditions to baseline identified from group-level hierarchical clustering, an additional distance matrix was constructed to determine the specific joints most impacted by WR loading. Each time-continuous joint angle was averaged across sprints and participants within each condition, and was used as a separate input variable. Data was scaled and a Euclidean distance dissimilarity measure was used to generate the distance matrix. A greater distance between the same joint and plane under different conditions was interpreted as greater dissimilarity in the specific motion of the joint. Code for the joint-level distance matrix analysis is provided at https://github.com/ktrounson/WR-acceleration/blob/main/Joint-distance.

## Results

### Descriptive statistics

Means and ranges of 20 m sprint time and whole body spatiotemporal measures for each participant and across the group are displayed in Table 2. Means and ranges of 10 m split times and COM velocity at 4 m are provided in S1 Table.

### Hierarchical agglomerative clustering

Group-level whole body coordination cluster analysis revealed that coordination in light WR conditions tended to be more similar to baseline, as indicated by lower branch heights from the baseline condition (Fig 4). The WR condition most similar to baseline was the light posterior shank condition. The two WR conditions most dissimilar to baseline were the heavy posterior thigh and heavy anterior thigh conditions.

Whole body coordination patterns of each individual were clustered together irrespective of the WR condition (Fig 5). Differences in coordination patterns between individuals were therefore greater than the changes to individual coordination induced by the addition of WR, highlighting the uniqueness of individual acceleration stride coordination. The extent of coordination dissimilarity induced by WR in general compared with baseline varied across participants. Responses to each WR magnitude and configuration also differed participant-to-participant. P2 demonstrated the most distinct coordination from the group and also exhibited relatively more similar coordination to baseline in the presence of WR in general, as indicated by lower average branch heights across conditions. P2 was the only participant for which the most similar coordination to baseline was expressed in the presence of a heavy WR condition (heavy anterior shank). For P3, coordination in the presence of light shank loading was very similar to baseline, while coordination in the heavy posterior thigh condition was markedly

**Table 2. Mean and range of 20 m sprint times and whole body spatiotemporal measures.**

| | | Baseline | HAT | LAT | HPT | LPT | HAS | LAS | HPS | LPS |
|---|---|---|---|---|---|---|---|---|---|---|
| 20 m sprint time (s) | P1 | 3.27 (3.21–3.33) | 3.32 (3.30–3.34) | 3.31 (3.27–3.35) | 3.35 (3.29–3.40) | 3.24 (3.18–3.30) | 3.32 (3.28–3.35) | 3.26 (3.21–3.29) | 3.29 (3.25–3.34) | 3.33 (3.28–3.38) |
| | P2 | 3.51 (3.49–3.55) | 3.57 (3.49–3.66) | 3.65 (3.60–3.69) | 3.62 (3.58–3.67) | 3.55 (3.51–3.58) | 3.60 (3.55–3.66) | 3.57 (3.56–3.59) | 3.56 (3.51–3.60) | 3.61 (3.60–3.63) |
| | P3 | 3.25 (3.21–3.34) | 3.32 (3.29–3.35) | 3.22 (3.20–3.24) | 3.19 (3.18–3.21) | 3.24 (3.20–3.26) | 3.29 (3.28–3.30) | 3.28 (3.21–3.35) | 3.33 (3.28–3.38) | 3.42 (3.37–3.50) |
| | P4 | 3.19 (3.15–3.23) | 3.26 (3.24–3.30) | 3.19 (3.17–3.21) | 3.19 (3.18–3.20) | 3.19 (3.18–3.20) | 3.18 (3.18–3.19) | 3.22 (3.18–3.26) | 3.23 (3.22–3.26) | 3.17 (3.17–3.18) |
| | P5 | 3.33 (3.29–3.39) | 3.30 (3.29–3.31) | 3.30 (3.29–3.32) | 3.40 (3.39–3.41) | 3.28 (3.27–3.29) | 3.29 (3.27–3.30) | 3.36 (3.35–3.36) | 3.23 (3.25–3.31) | 3.26 (3.22–3.31) |
| | Group | 3.31 (3.15–3.55) | 3.36 (3.24–3.66) | 3.34 (3.17–3.69) | 3.35 (3.18–3.67) | 3.30 (3.18–3.58) | 3.34 (3.18–3.66) | 3.34 (3.18–3.59) | 3.34 (3.22–3.60) | 3.36 (3.17–3.63) |
| Flight time (ms) | P1 | 85 (80–92) | 100 (92–104) | 80 (68–100) | 88 (72–108) | 83 (76–92) | 92 (88–96) | 85 (76–92) | 77 (76–80) | 84 (72–104) |
| | P2 | 83 (72–88) | 88 (80–96) | 79 (76–84) | 76 (72–80) | 80 (76–84) | 88 (84–92) | 85 (80–88) | 81 (72–88) | 79 (76–84) |
| | P3 | 81 (72–92) | 91 (88–92) | 81 (80–84) | 83 (80–88) | 76 (68–88) | 84 (80–88) | 80 (80–88) | 92 (84–96) | 81 (76–84) |
| | P4 | 81 (76–92) | 80 (80–88) | 84 (80–88) | 85 (84–88) | 84 (76–88) | 83 (76–88) | 81 (72–96) | 83 (80–84) | 77 (76–80) |
| | P5 | 93 (88–100) | 104 (96–112) | 84 (76–96) | 88 (84–92) | 99 (80–120) | 84 (76–88) | 76 (72–80) | 95 (88–100) | 96 (92–100) |
| | Group | 85 (72–100) | 93 (80–112) | 82 (68–100) | 84 (72–108) | 84 (68–120) | 86 (76–96) | 82 (72–96) | 86 (72–100) | 83 (72–100) |
| Ground contact time (ms) | P1 | 136 (132–140) | 136 (132–140) | 151 (144–156) | 145 (136–152) | 140 (136–144) | 133 (128–140) | 137 (136–140) | 140 (136–144) | 147 (144–152) |
| | P2 | 165 (160–172) | 167 (164–168) | 171 (168–172) | 177 (172–180) | 169 (168–172) | 169 (164–172) | 164 (160–168) | 172 (164–180) | 173 (172–176) |
| | P3 | 167 (160–176) | 167 (160–172) | 175 (172–176) | 164 (160–168) | 169 (160–180) | 168 (164–172) | 167 (152–180) | 169 (164–176) | 179 (176–180) |
| | P4 | 131 (124–136) | 131 (120–140) | 131 (124–136) | 125 (124–128) | 128 (120–132) | 128 (124–132) | 135 (124–144) | 123 (120–128) | 129 (128–132) |
| | P5 | 156 (152–160) | 149 (144–156) | 165 (156–176) | 163 (156–172) | 161 (152–168) | 173 (168–180) | 172 (168–176) | 160 (156–168) | 158 (156–160) |
| | Group | 151 (124–176) | 150 (120–172) | 158 (124–176) | 155 (124–180) | 154 (120–180) | 154 (124–180) | 155 (124–180) | 153 (120–180) | 157 (128–180) |
| Step length (m) | P1 | 1.34 (1.31–1.37) | 1.42 (1.36–1.45) | 1.36 (1.28–1.46) | 1.37 (1.34–1.40) | 1.33 (1.30–1.38) | 1.36 (1.34–1.39) | 1.39 (1.36–1.42) | 1.32 (1.28–1.34) | 1.35 (1.27–1.47) |
| | P2 | 1.40 (1.39–1.41) | 1.42 (1.39–1.44) | 1.39 (1.36–1.41) | 1.40 (1.36–1.45) | 1.38 (1.35–1.41) | 1.45 (1.44–1.45) | 1.39 (1.39–1.40) | 1.40 (1.39–1.42) | 1.44 (1.43–1.45) |
| | P3 | 1.49 (1.46–1.52) | 1.50 (1.49–1.52) | 1.54 (1.48–1.54) | 1.51 (1.48–1.54) | 1.49 (1.46–1.50) | 1.51 (1.49–1.52) | 1.50 (1.48–1.54) | 1.52 (1.49–1.56) | 1.54 (1.52–1.57) |
| | P4 | 1.27 (1.23–1.33) | 1.23 (1.17–1.28) | 1.27 (1.26–1.28) | 1.24 (1.23–1.25) | 1.26 (1.24–1.27) | 1.28 (1.26–1.29) | 1.28 (1.22–1.34) | 1.24 (1.21–1.26) | 1.24 (1.24–1.25) |
| | P5 | 1.42 (1.34–1.47) | 1.50 (1.46–1.53) | 1.36 (1.33–1.38) | 1.47 (1.43–1.52) | 1.58 (1.53–1.63) | 1.42 (1.40–1.44) | 1.38 (1.33–1.40) | 1.46 (1.38–1.50) | 1.54 (1.54–1.55) |
| | Group | 1.38 (1.23–1.52) | 1.41 (1.17–1.53) | 1.39 (1.26–1.54) | 1.40 (1.23–1.54) | 1.41 (1.24–1.63) | 1.40 (1.26–1.52) | 1.39 (1.22–1.54) | 1.39 (1.21–1.56) | 1.41 (1.24–1.57) |
| Strep frequency (Hz) | P1 | 4.42 (4.31–4.55) | 3.99 (3.97–4.03) | 4.35 (4.03–4.55) | 4.15 (3.97–4.24) | 4.50 (4.24–4.72) | 4.27 (4.17–4.46) | 4.26 (4.17–4.39) | 4.49 (4.39–4.55) | 4.37 (3.85–4.81) |
| | P2 | 3.87 (3.79–3.97) | 3.81 (3.73–3.91) | 3.77 (3.73–3.85) | 3.79 (3.73–3.85) | 3.89 (3.79–4.03) | 3.68 (3.62–3.73) | 3.97 (3.91–4.03) | 3.83 (3.79–3.85) | 3.83 (3.73–3.91) |
| | P3 | 3.90 (3.73–4.03) | 3.70 (3.57–3.91) | 3.93 (3.79–4.10) | 3.91 (3.85–4.03) | 4.04 (3.85–4.24) | 3.77 (3.73–3.79) | 3.87 (3.85–3.91) | 3.77 (3.73–3.79) | 3.72 (3.62–3.85) |
| | P4 | 4.48 (4.24–4.72) | 4.34 (4.31–4.39) | 4.60 (4.46–4.72) | 4.49 (4.46–4.55) | 4.41 (4.31–4.55) | 4.55 (4.55–4.56) | 4.40 (4.03–4.72) | 4.47 (4.39–4.55) | 4.57 (4.46–4.63) |
| | P5 | 3.80 (3.70–3.97) | 3.73 (3.62–3.79) | 3.79 (3.62–3.91) | 3.83 (3.73–3.91) | 3.74 (3.47–3.97) | 3.89 (3.85–3.91) | 3.95 (3.85–4.03) | 3.83 (3.79–3.85) | 3.88 (3.79–3.97) |
| | Group | 4.11 (3.70–4.72) | 3.91 (3.57–4.39) | 4.09 (3.62–4.72) | 4.03 (3.73–4.55) | 4.12 (3.47–4.72) | 4.03 (3.62–4.56) | 4.09 (3.85–4.72) | 4.08 (3.73–4.55) | 4.09 (3.62–4.81) |

HAT, heavy anterior thigh; LAT, light anterior thigh; HPT, heavy posterior thigh; HAS, heavy anterior shank; LAS, light anterior shank; LPS, light posterior shank; HPS, heavy posterior shank; LPT, light posterior thigh; COM, centre of mass.

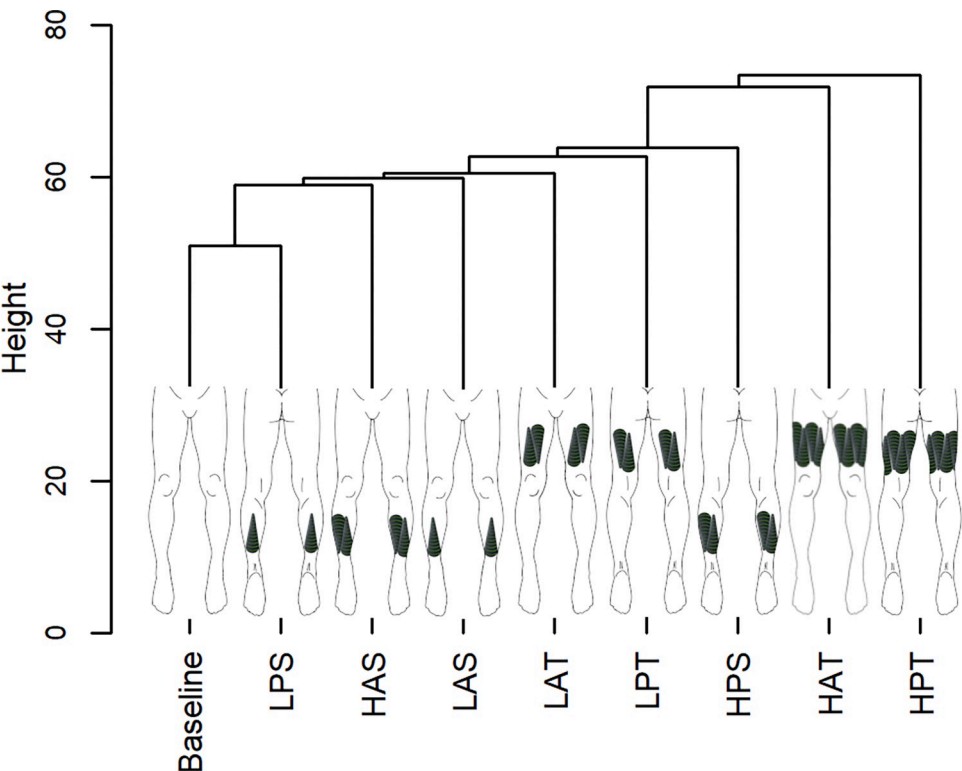

**Fig 4. Hierarchical cluster analysis of whole body coordination at group-level.** The height of branches indicates the degree of dissimilarity between coordination patterns derived from the Euclidean distance measure. Diagrammatic representation of each WR condition is included on the relevant branch. LPS, light posterior shank; HAS, heavy anterior shank; LAS, light anterior shank; LAT, light anterior thigh; LPT, light posterior thigh; HPS, heavy posterior shank; HAT, heavy anterior thigh; HPT, heavy posterior thigh.

different. The most substantial within-individual deviation of coordination from baseline was shown by P1 in the presence of heavy anterior thigh WR.

## Self-organising map

Fig 6 presents the trained SOM and best-matching unit trajectories for each participant in the two most dissimilar (heavy posterior thigh and heavy anterior thigh) conditions to baseline. Best-matching unit trajectories for each participant in the remaining conditions are provided in S1 Fig. While the magnitude of change to coordination brought about by heavy anterior and heavy posterior thigh WR appeared similar in P4 and P5 according to the within-individual hierarchical cluster analysis, the best-matching unit trajectories reveal that the characteristics of the coordinative changes were different with respect to the portion of the stride cycle affected. Looking between each touchdown and toe-off event, the best-matching unit trajectories for each participant can be compared between conditions to understand where in the stride cycle differences manifested. For P4, the entire stride appeared to be affected in the heavy posterior thigh condition, whereas the middle portion of the stride appeared most affected in the heavy anterior thigh condition. For P5, the period between left foot toe-off and right foot toe-off appeared most affected, particularly in the heavy posterior thigh condition. P1 exhibited markedly dissimilar coordination in the heavy anterior thigh condition compared with baseline between first left foot toe-off to right foot toe-off. Lastly, for P3, the end of the stride cycle between left foot touchdown and the second instance of left foot toe-off differed substantially relative to baseline in the heavy posterior thigh condition.

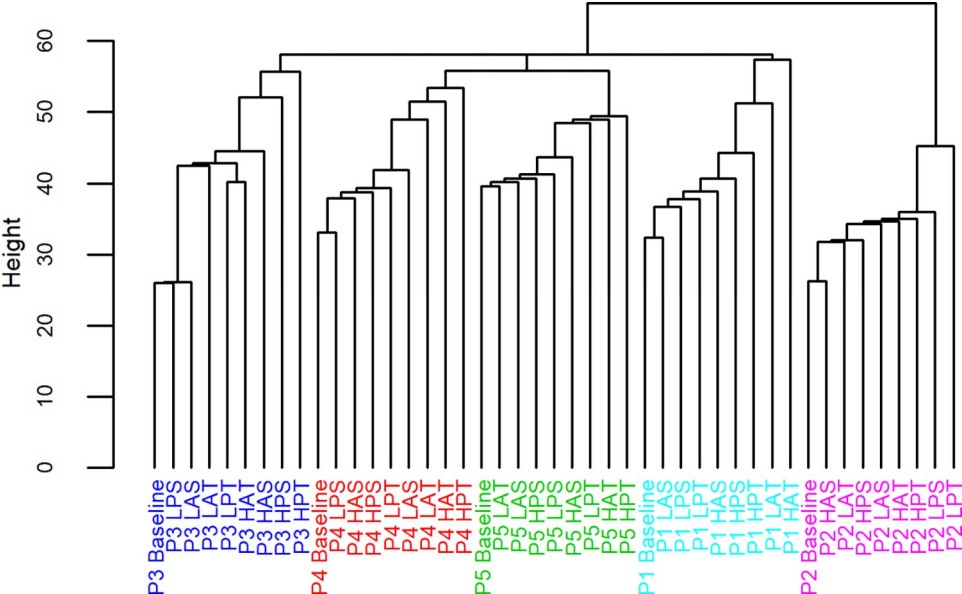

**Fig 5. Hierarchical cluster analysis of whole body coordination within-individuals.** The height of branches indicates the degree of dissimilarity between coordination patterns derived from the Euclidean distance measure. Unique colouring is used in addition to participant initials to distinguish between individuals. LPS, light posterior shank; HAS, heavy anterior shank; LAS, light anterior shank; LAT, light anterior thigh; LPT, light posterior thigh; HPS, heavy posterior shank; HAT, heavy anterior thigh; HPT, heavy posterior thigh.

## Joint-level distance matrix

The results of the distance matrices constructed from time-continuous joint angles between baseline and the two most dissimilar conditions (heavy posterior thigh and heavy anterior thigh) are presented in S2 Table. For baseline versus the heavy posterior thigh condition, the five most dissimilar joints and motion planes were the pelvis segment in the sagittal plane, right shoulder in the transverse plane, right and left hips in the sagittal plane, and right shoulder in the sagittal plane. For baseline versus the heavy anterior thigh condition, the five most dissimilar joints and motion planes were the right and left shoulders in the transverse plane, thorax in the sagittal plane, and right and left shoulders in the sagittal plane. Time series ensemble means ± SD for these joints and planes are presented in Fig 7. On average, in the heavy posterior thigh condition, pelvic orientation was closer to upright standing and there was greater hip extension throughout the stride cycle on both the left and right side compared with baseline. In the heavy anterior thigh condition, amplitude of movement at the shoulders in the transverse and sagittal planes appeared greater compared with baseline.

## Discussion

### Main findings

This study sought to investigate the extent and manner of changes to whole body coordination during early acceleration in response to the addition of various WR loading configurations and magnitudes among five Australian Rules football players. Across the participant group, heavy posterior and anterior thigh WR conditions brought about the most dissimilar coordination patterns compared with baseline. On average, heavy posterior thigh WR loading resulted in a more neutral pelvic position with greater hip extension throughout the stride cycle, while coordination dissimilarity in the presence of heavy anterior thigh loading

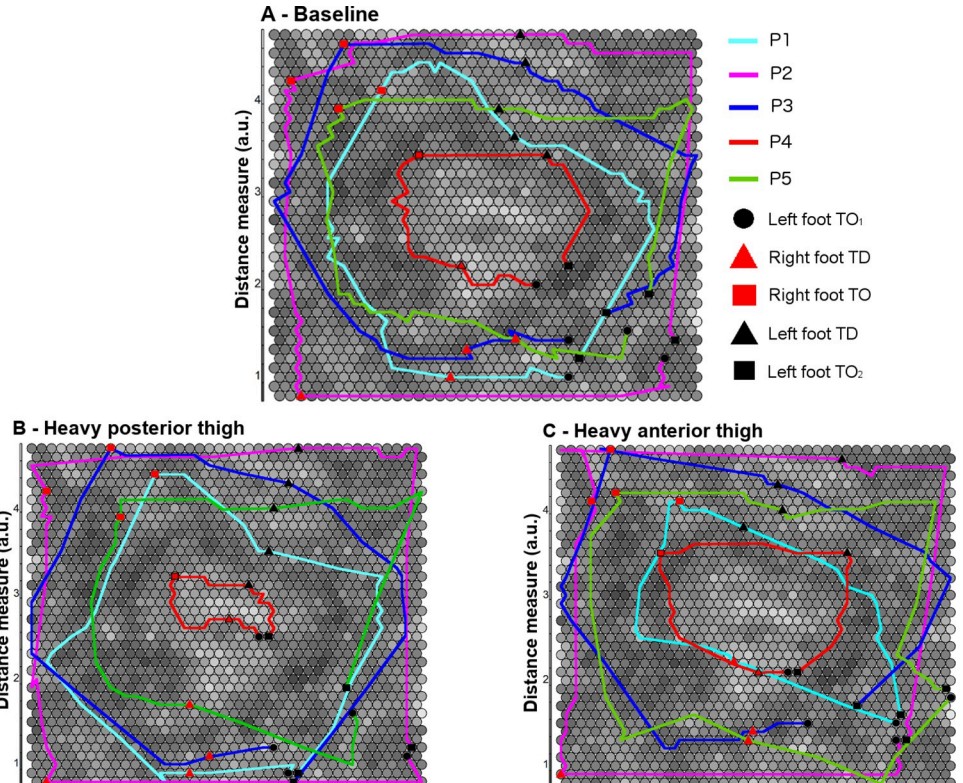

**Fig 6. Trained SOM and best-matching unit trajectories for each participant.** (A) Baseline. (B) Heavy posterior thigh condition. (C) Heavy anterior thigh condition. Shading indicates the distance of each cell to its neighbours with darker shaded areas having a smaller distance. Participants are indicated by unique colours. Shapes and colours are used to indicate key phases of the stride cycle. Black circle, first left foot toe-off (beginning of stride) ($TO_1$); red triangle, right foot touchdown (TD); red square, right foot toe-off (TO); black triangle, left foot touchdown (TD); black square, second left foot toe-off (end of stride) ($TO_2$).

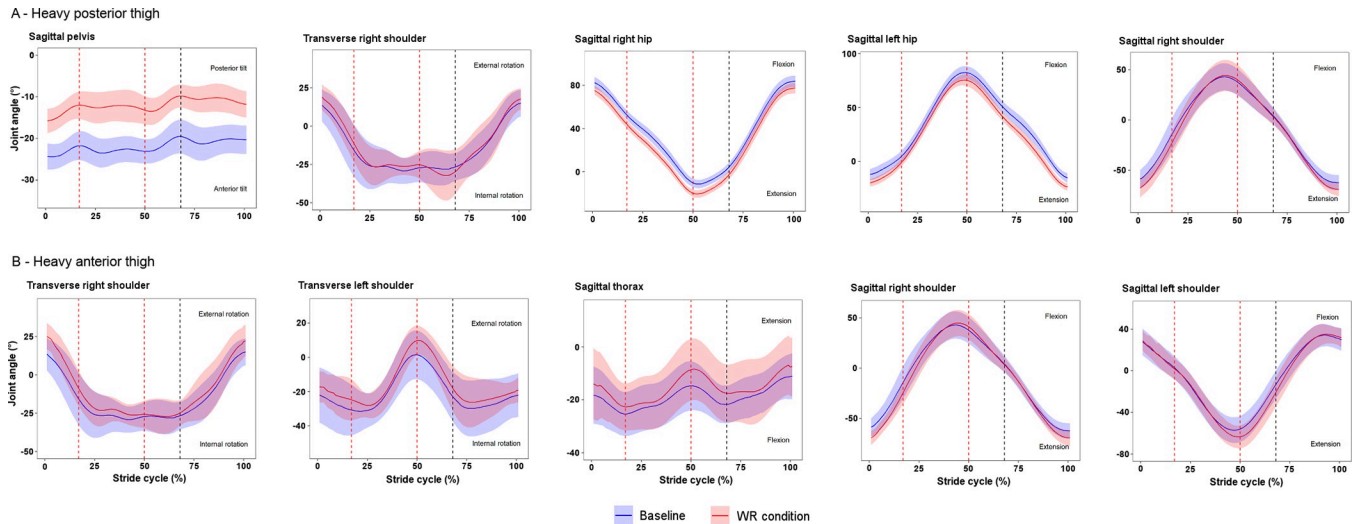

**Fig 7. Joint angle means ± SD for most dissimilar joints and motion planes throughout stride cycle as indicated by joint-level distance matrix.** (A) Baseline versus heavy posterior thigh condition. (B) Baseline versus heavy anterior thigh condition. Dark lines indicate ensemble means and shaded areas indicate SD. Vertical dashed lines represent touchdown and toe-off on the right foot (red) and touchdown on the left foot (black).

manifested most in upper body joints, particularly the shoulders in the transverse and sagittal planes. Coordination was most similar to baseline in the light posterior shank WR condition.

The findings offer a starting point for coaches seeking to use WR as a movement control parameter to alter acceleration technique. Coaches wanting to promote greater hip extension or increase the upper body contribution to acceleration among their athletes, for example, may start by exploring posterior and anterior thigh WR, respectively. Alternatively, coaches may use WR to create movement variability generally as a means to encourage autonomous exploration of different coordinative states [65, 66]. This can facilitate athlete-driven technique changes and improved performance, particularly if interspersed with unloaded sprints and coupled with knowledge of results (i.e. sprint time) [67–69]. The findings of this study suggest that exposure to relatively heavier WR loading magnitudes may be more appropriate for this purpose.

Importantly, individual-level analyses in the form of hierarchical agglomerative clustering and self-organising maps demonstrated that each participant had a clearly distinct coordination pattern. Participants also differed in the extent and manner in which their coordination changed when WR was applied. Although heavy thigh WR conditions tended to alter coordination to a greater extent, this was not uniformly the case. Coaches must keep this in mind when pursuing the use of WR, or any constraint, as a movement control parameter, especially in the context of a team setting.

## Comparisons to previous WR research

As a matter of situating this study within existing WR research it is important to note that average acceleration outcome measures (20 m sprint time, 10 m split time, and COM velocity at 4 m) differed minimally based on WR condition across the group. There were also no obvious trends in relation to whole body spatiotemporal measures. While inferential analyses were not performed on these data, similar findings in larger participant cohorts have been reported with lower body WR of comparable magnitudes [32, 38]. Other studies have described decreased stride frequency and increased ground contact time, though the minimal change to early acceleration performance appears consistent [40, 41]. Maintenance of acceleration in the presence of a WR constraint appears to suggest exploitation of movement system degeneracy among the group. The heavy posterior thigh condition, for example, which gave rise to the most dissimilar coordination patterns compared with baseline, only increased 20 m sprint time by 0.04 s on average.

## Skill acquisition and coaching implications

For field-based sport athletes, there is a need for acceleration coordination patterns to be adaptable given the vast array of unique scenarios that can emerge from the interactions between task (e.g. evading an opponent), environmental (e.g. slippery playing surface), and organismic (e.g. fatigue) constraints [70, 71]. In practical terms, movement adaptability serves to limit performance outcome variability, and is a hallmark of higher performing athletes across many sports [72, 73]. Increased adaptability may also attenuate injury risk, especially in the context of actions performed repeatedly, by distributing stress across a wider variety of structures [74–76]. Critically, the capacity for individuals to be adaptable appears to be trainable through exposure to novel constraints, driving exploration of alternate coordination patterns capable of maintaining task performance [48, 73]. In this study, the most similar coordination pattern to a baseline condition as indicated by individual-level hierarchical clustering was exhibited by P3 in the light posterior shank condition. Surprisingly, this was also the condition in which the greatest average increase in 20 m sprint time occurred (+0.17 s from baseline). In this instance, the WR may have been an insufficient perturbation to move

the participant away from their stable baseline coordination and/or the participant may not have perceived a diminution in acceleration so did not seek an alternate movement strategy to maintain task performance [77]. This is therefore likely a less effective loading configuration and magnitude for training movement adaptability in this individual. In contrast, the most dissimilar coordination pattern to a baseline condition was exhibited by P1 in the heavy anterior thigh condition, and was accompanied by only a slight increase in 20 m sprint time (+0.05 s from baseline). This suggests a suitable challenge to sprint adaptability for this individual. Training studies with pre- and post-training coordination variability assessments are needed, however, to make definitive, generalisable conclusions in this respect.

Despite subtle within-individual differences, participants generally exhibited the most dissimilar coordination in heavy thigh WR conditions compared with baseline sprints. P2 was a notable exception to this trend, however, showing no clear pattern in the responses to WR. This participant also had the most distinct coordination and slowest 20 m sprint times, suggesting that sprint acceleration skill level may have been lower than other participants. Athlete skill level is yet another practical consideration for coaches. Lesser skilled individuals often have higher coordination variability generally [78]. This could explain the less predictable responses to WR in P2. For such individuals, repetitive practice with minimal alteration to constraints may be preferable [79].

WR may be a suitable constraint to channel coordination patterns toward organisational states deemed favourable for performance [35, 42]. Among this participant group, heavy thigh WR loading effected the greatest change in whole body coordination compared with baseline acceleration. It is not obvious why heavy thigh loading brought about greater changes than heavy shank loading, though it may have been due to the greater system load in the former condition. This difference was a necessary consequence of the decision to match thigh and shank loads on the basis of changes to the moment of inertia about the hip throughout the acceleration stride. For the heavy posterior thigh condition, coordination shifted towards the adoption of greater hip extension throughout the stride and a more neutral pelvic position. The tendency toward greater hip extension may have been an effect of the posterior shift in mass of the thigh segments. Pelvic position in the heavy posterior thigh condition may have changed to maintain the preferred relationship between the global centre of mass and the posteriorly-shifted base of support arising from greater peak hip extension [80]. Given the importance of hip extension for propulsion during acceleration, posterior thigh WR has potential as a coaching tool to accentuate this motion among athletes for whom this is identified as a technical shortcoming [81, 82]. Future research should focus specifically on the effect of this loading scheme on sagittal plane pelvis and hip kinematics among a larger athlete sample to verify the generalisability of such a prescription.

In terms of heavy anterior thigh WR, movement amplitude at the shoulders appeared to increase on average in both the sagittal and transverse planes. Though the role of arms during in acceleration is debated [83], there is clearly high movement coupling between each shoulder and contralateral hip joint [84]. With heavy anterior thigh WR loading, the increased arm angular displacement may have acted to preserve proportionality between the relative rotational work performed at the shoulders and hips [46, 85]. Heavy anterior thigh WR could therefore serve as a constraint to promote arm swing action during acceleration, though consideration must be given to whether a given athlete is likely to benefit from accentuated movement in both the sagittal and transverse planes.

## Limitations

As addressed throughout, a limitation of this study is the small sample size. This type of exploratory study does, however, offer starting points for coaches working in applied settings and

important signposts for future investigations. It should also be reiterated that the findings pertain only to a specific portion of early sprint acceleration (steps 3 and 4) and that the effects of WR likely differ in other phases of sprinting, such as at maximum velocity. Worth noting also is that complex systems-based pedagogical approaches emphasise ecological validity in training. Ideally, this would extend to the testing environment also, though the acquisition of detailed kinematic data in on-field settings poses challenges. Lastly, individuals may have naturally exhibited small improvements in sprint acceleration performance over the testing period as a function of high frequency sprint exposures during match play over the course of their competitive season [86, 87].

## Conclusions

Across the participant group, heavy WR applied to the thighs had the greatest effect on whole body coordination during sprint acceleration. On average, heavy posterior thigh WR led to altered pelvic position and greater hip extension, while heavy anterior thigh WR led to accentuated movement at the shoulders in the transverse and sagittal planes. Future research may investigate these specific effects in a larger sample group to determine the generalisability of findings. Given the absence of other research into the changes to whole body coordination induced by WR, heavy thigh WR may be an appropriate starting point for coaches seeking to use WR as a movement control parameter or as a tool to promote movement variability in acceleration for field-based sport athletes. Coaches should note, however, that individuals did exhibit variation in the extent and manner in which each WR condition altered coordination, which may have been the result of differences in individual coordination dynamics and/or skill level.

## Supporting information

**S1 Table. Mean and range of 10 m split times and COM velocity at 4 m mark.**
(DOCX)

**S2 Table. Euclidian distance dissimilarity measures of joint angles between baseline and the heavy posterior thigh and heavy anterior thigh conditions.**
(DOCX)

**S1 Fig. Trained SOM and best-matching unit trajectories for each participant.** (A) Heavy posterior shank condition. (B) Heavy anterior shank condition. (C) Light posterior thigh condition. (D) Light anterior thigh condition. (E) Light posterior shank condition. (F) Light anterior shank condition. Participants are indicated by unique colours (light blue, P1; magenta, P2; dark blue, P3; red, P4; green, P5). Shapes and colours are used to indicate key phases of the stride cycle. Black circle, first left foot toe-off (beginning of stride) ($TO_1$); red triangle, right foot touchdown (TD); red square, right foot toe-off (TO); black triangle, left foot touchdown (TD); black square, second left foot toe-off (end of stride) ($TO_2$).
(TIFF)

## Acknowledgments

The authors would like to acknowledge the research participants for their involvement in this study.

## Author Contributions

**Conceptualization:** Karl M. Trounson, Sam Robertson.

**Data curation:** Karl M. Trounson.

**Formal analysis:** Karl M. Trounson.

**Investigation:** Karl M. Trounson.

**Methodology:** Karl M. Trounson.

**Supervision:** Sam Robertson, Kevin Ball.

**Visualization:** Karl M. Trounson.

**Writing – original draft:** Karl M. Trounson.

**Writing – review & editing:** Sam Robertson, Kevin Ball.

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
