## [Decision Letter · Decision Letter 0]

8 Feb 2024

PONE-D-23-30684The influence of lightweight wearable resistance on whole body coordination during sprint acceleration among Australian Rules football players.PLOS ONE

Dear Dr. Trounson,

Thank you for submitting your manuscript to PLOS ONE. After careful consideration, we feel that it has merit but does not fully meet PLOS ONE’s publication criteria as it currently stands. Therefore, we invite you to submit a revised version of the manuscript that addresses the points raised during the review process.

We look forward to receiving your revised manuscript.

Kind regards,

Ersan Arslan, Ph.D.

Academic Editor

PLOS ONE

3. We note that Figure 1, 2, 3 and 4 in your submission contain copyrighted images. All PLOS content is published under the Creative Commons Attribution License (CC BY 4.0), which means that the manuscript, images, and Supporting Information files will be freely available online, and any third party is permitted to access, download, copy, distribute, and use these materials in any way, even commercially, with proper attribution. For more information, see our copyright guidelines: http://journals.plos.org/plosone/s/licenses-and-copyright.

a. You may seek permission from the original copyright holder of Figure 1, 2, 3 and 4 to publish the content specifically under the CC BY 4.0 license. 

Reviewers' comments:

Reviewer's Responses to Questions

**Comments to the Author**

1. Is the manuscript technically sound, and do the data support the conclusions?

Reviewer #1: No

Reviewer #2: Yes

2. Has the statistical analysis been performed appropriately and rigorously? 

Reviewer #1: No

Reviewer #2: Yes

3. Have the authors made all data underlying the findings in their manuscript fully available?

Reviewer #1: Yes

Reviewer #2: Yes

4. Is the manuscript presented in an intelligible fashion and written in standard English?

Reviewer #1: No

Reviewer #2: Yes

5. Review Comments to the Author

Reviewer #1: Dear Authors,

I would like to express my gratitude for the opportunity to review this manuscript.

The manuscript at this stage requires improvements. Below are suggestions with line indications:

1-3 – Please revise the title, author´s name, and affiliation format, considering the journal template and instructions for authors.

26 – Please write in full all abbreviations in the first appearance in the manuscript.

21-45 – The abstract is too long, and results are missing. Moreover, no keywords are observed. Please revise.

52 – Please revise the citation format.

54 – Please revise the citations format (space between numbers). Please consider this in all manuscript.

63 – Too many citations. Revision is suggested.

73-99 – Please consider shorter paragraphs in all manuscript to increase readability.

100 – Please correct the citation format. Same in line 243 (please revise all manuscript).

Near 129 – The research gap and study aim should be clearly presented at the end of the discussion section.

137 – Please indicate inclusion and exclusion criteria and describe the sample (performance level, training routines, and others).

142 – Please indicate the number of code approval.

148 – Please indicate all procedures. Time of data collection? Circadian effect? Lab conditions (space, temperature, humidity). All procedures should be clearly understood by the readers.

222 – At the end of this section all details related to the methodology should be clearly understood by the readers. Please confirm.

271 – Please consider reorganizing this section aiming for more direct and clear information to the readers. For example, a sample of 5 is not mentioned. Sample Power (Gpower)? Additionally, the statistical analysis requires a more depth analysis, one M ± SD is observable.

356 – Please consider the journal template and instructions for authors in the format of the table. Moreover, please revise the table content.

348-455 – Please consider a different format in the results section. Too many text difficult the interpretation.

455-594 - Please consider a different format in the discussion section (for example with subsections). Too many text difficult the interpretation.

Near 594 - Please include suggestions for future research.

538 - All references should be carefully revised, they are not according to the journal template and instructions for authors.

598-610 – Please consider more direct messages in the conclusions and end of the abstract. If possible, with practical application.

612 – More information is requested besides acknowledgments.

616 - Please revise the format of the manuscript considering the journal template and instructions for authors. Please check all details.

Please carefully revise English details throughout the manuscript.

Pages 43-49 – Please improve the figures quality. Please include the units in the axis.

Reviewer #2: I thank the Authors and the Editors for the chance to review the present article. The manuscript investigates the influence of lightweight wearable resistance on whole-body coordination during sprints in Australian Rules football players. The manuscript is well structured and written, and I compliment the Authors for designing such a remarkable analysis. There are, however, some criticisms that should be addressed before the article can be deemed suitable for publication.

The introduction is informative and comprehensive about the topic. However, some paragraphs seem to be redundant, with the risk of making the section wordy and complicated to follow for the reader (e.g., lines 73-99). I would suggest streamlining the section and focusing more on the study rationale, which seems to be vague at this point. The purpose appears more in the direction of highlighting the possibility of the presented analysis framework being implemented in future larger studies rather than revealing possible effects in the presented dataset. It should be clarified whether the presented manuscript should be considered an original study, even if no inferential statistics have been conducted, from which to draw some conclusions or a framework to develop future studies.

Similar to the introduction section, the discussion is way too extensive, and it seems to point the attention more on being a foundation for future studies (lines 460 and 469) rather than on the actual findings. I would suggest reorganizing the section, focusing more on the effects of WR. On a related note, the number of references (more than 100) seems excessive for a cross-sectional original study involving a small sample size; I would suggest reducing them in order to facilitate the readers to be informed on the topic.

Minor comments:

Line 24: WR should be defined here as an acronym.

Line 58: a brief definition of the factors most influencing sprint acceleration could be helpful.

Line 64: please define the concept of “technique interventions”.

Lines 112-114: please clarify this notion.

Line 135: what is the usual weekly/monthly training load of the involved athletes? What is their Australian Rules football experience?

Lines 250 and 261: how were kinematics (CoM and angles) computed in terms of software/coding/modeling?

Lines 492-494: were inferential statistics used to study these differences?

6. PLOS authors have the option to publish the peer review history of their article (what does this mean?). If published, this will include your full peer review and any attached files.

Reviewer #1: No

Reviewer #2: No

---

## [Author Response · Author response to Decision Letter 0]

13 Apr 2024

Response to editor’s comments:

We amended the file names of all figures to exclude spaces, such that these now read “Fig1.tif”.

We amended table titles to bold type throughout the manuscript.

Symbols for contributorship are now included on the title page.

Font and line spacing in references section has been amended according to style requirements.

We hope that these changes satisfy the style requirements as described in the referred templates.

An Ethics statement is included in the “Methods” section under the sub-section entitled “Participants”. Excerpts from the original manuscript read, “All [participants] provided written informed consent…All procedures used in this study complied with the criteria of the declaration of Helsinki and the ethical approval [was] granted by the Victoria University Human Research Ethics Committee.”

This statement includes the full name of the ethics committee who approved the study and states that written consent was obtained from participants.

Please advise if a separate statement under a distinct sub-section is required.

3. We note that Figure 1, 2, 3 and 4 in your submission contain copyrighted images. All PLOS content is published under the Creative Commons Attribution License (CC BY 4.0), which means that the manuscript, images, and Supporting Information files will be freely available online, and any third party is permitted to access, download, copy, distribute, and use these materials in any way, even commercially, with proper attribution. For more information, see our copyright guidelines: http://journals.plos.org/plosone/s/licenses-and-copyright.

a. You may seek permission from the original copyright holder of Figure 1, 2, 3 and 4 to publish the content specifically under the CC BY 4.0 license.

Figure 1 and 2 were originally produced by lead author Karl Trounson and have been previously published in PLoS ONE. As such, a request for permission form signed by Karl Trounson is now included in the submission, and the following attribution is provided in figure legends: “Reprinted from Trounson KM, Busch A, French Collier N, Robertson S (2020) Effects of acute wearable resistance loading on overground running lower body kinematics. PLoS ONE 15(12): e0244361 under a CC BY license, with permission from PLoS ONE, original copyright 2020.”

Figure 3 and 4 have been amended using an outline available in the public domain.

Responses to reviewer #1:

1-3 – Please revise the title, author´s name, and affiliation format, considering the journal template and instructions for authors.

Symbols for author contributorship are now included on the title page in line with formatting guidelines. Besides this, formatting of the title, authors’ names, and affiliations in the original manuscript appear to adhere to the journal requirements as set forth in the affiliations formatting guidelines: (https://journals.plos.org/plosone/

s/file?id=3fac/PLOS Affiliations Formatting Guidelines.pdf)

26 – Please write in full all abbreviations in the first appearance in the manuscript.

We have amended abbreviations throughout the manuscript in line with this feedback such that all are written out in full at first appearance.

21-45 – The abstract is too long, and results are missing. Moreover, no keywords are observed. Please revise.

The abstract in the original manuscript does not exceed 300 words, keeping in line with PLoS ONE submission guidelines. Data analysis in this study takes the form of cluster analysis, self-organising maps, and distance matrices, the results of which are described in lines 38-43 of the revised abstract. No inferential statistics were performed given the small sample size, which is why numerical data aren’t reported here.

There is no stipulation in submission guidelines to include keywords in the manuscript. These are provided during the submission process and can be seen on the first page of the submission file.

52 – Please revise the citation format.

PLoS citation style file for Endnote (available at: https://endnote.com/downloads/

styles/plos-public-library-of-science-all-journals/) was used for all references in the original manuscript. As such, all references should, and indeed appear to, adhere to the PLoS style requirements.

54 – Please revise the citations format (space between numbers). Please consider this in all manuscript.

Spacing between numbers if two consecutive citations are included appears to be the standard for PLoS articles. Again, the PLoS citation style file was used for referencing throughout the manuscript, and we understand that this is consistent with journal requirements.

63 – Too many citations. Revision is suggested.

Edits have been made throughout the manuscript to reduce the number of citations and omit less relevant information in order to keep the key points of the paper more clear. The number of citations is substantially reduced in the revised manuscript.

73-99 – Please consider shorter paragraphs in all manuscript to increase readability.

We acknowledge that lengthy sentences and paragraphs affect readability of the original manuscript and have made attempts to shorten these throughout in the revised manuscript. Much of the section identified by the reviewer (73-99) has been omitted or rewritten to improve clarity.

100 – Please correct the citation format. Same in line 243 (please revise all manuscript). 

Citation formatting adheres to the PLoS citation style, in line with previous comment responses. In the specific line identified, the paper author is explicitly named in the text. We were unable to find formatting instructions for this type of instance, however, we have followed the format demonstrated in a recently published PLoS paper (https://journals.plos.org/plosone/article?id=10.1371/journal.pone.0296860 - introduction section; paragraph 2; line 1).

Near 129 – The research gap and study aim should be clearly presented at the end of the discussion section.

The research gap and aim of the study are described in the final paragraph of the introduction. Edits have been made to this section to make the existing research gap into wearable resistance and the aim of the study clearer.

137 – Please indicate inclusion and exclusion criteria and describe the sample (performance level, training routines, and others).

We have made amendments to more clearly state inclusion and exclusion criteria. In addition, further information about typical Australian Rules football training habits of participants has been provided in the form of session frequency and typical running loads across a week.

142 – Please indicate the number of code approval.

Victoria University Human Research Ethics Committee code approval number (HRE19-020) has now been included in this section of the manuscript.

148 – Please indicate all procedures. Time of data collection? Circadian effect? Lab conditions (space, temperature, humidity). All procedures should be clearly understood by the readers.

Edits to the study design section have been made to include information relating to the time of data collection and laboratory conditions, as suggested.

222 – At the end of this section all details related to the methodology should be clearly understood by the readers. Please confirm.

Attempts have been made to clarify the details of the methodology in line with reviewer suggestions (e.g. laboratory conditions, participant training characteristics, testing session timing, and computation of joint kinematics). We recognise that details of data processing and data analysis may be challenging for readers without a biomechanics background, however, we have aimed to include as much detail as possible in order for the experimental setup and analyses to be reproduced.

271 – Please consider reorganizing this section aiming for more direct and clear information to the readers. For example, a sample of 5 is not mentioned. Sample Power (Gpower)? Additionally, the statistical analysis requires a more depth analysis, one M ± SD is observable.

This section has been amended to include the sample number included in the group average. The non-use of inferential statistics is emphasised throughout the manuscript and as such a power calculation is not relevant. The descriptive statistics form only a minor portion of the data analysis and are primarily reported for comparison with existing literature on the effects of WR during sprint acceleration. Hierarchical agglomerative clustering and self-organising maps serve as the main substance of the data analysis and we feel strongly that these are far more insightful analyses in relation to characterisation of whole body coordination changes in response to WR.

356 – Please consider the journal template and instructions for authors in the format of the table. Moreover, please revise the table content.

The formatting of the table has been corrected and the content revised to include a single main performance measure (20 m sprint time). 10 m split time and centre of mass velocity at 4 m have been moved to a supplementary table (S1 Table) to aid with interpretation and clarity of Table 2.

348-455 – Please consider a different format in the results section. Too many text difficult the interpretation.

The discussion has been restructured to aid with interpretation (see below), however, the results section is in convention with normal formatting, so this has been largely retained. We appreciate that there may be more text than normal, however, we feel that this is necessary in pointing the reader to pertinent elements of the hierarchical clustering dendrogram and self-organising map figures, which are the key elements of the results section.

455-594 - Please consider a different format in the discussion section (for example with subsections). Too many text difficult the interpretation.

Subsections have now been included in the discussion to aid interpretation and readability. In addition, the text has been shortened by removing unnecessary depth of commentary into proposed mechanisms of the observations. We hope that the revised version is more readable and interpretable and that key messages come across more clearly.

Near 594 - Please include suggestions for future research.

We suggest that future research should focus specifically on the effect of heavy anterior and posterior thigh wearable resistance on sagittal plane pelvis and hip kinematics among a larger athlete sample (lines 564 and 592).

538 - All references should be carefully revised, they are not according to the journal template and instructions for authors.

No reference was included in the line identified (538). Again, we would dispute that references are not formatted according to the journal template and instructions for authors, since the PLoS citation style file for Endnote was specifically used. We welcome specific examples from the reviewer in which we have deviated from the required citation style.

598-610 – Please consider more direct messages in the conclusions and end of the abstract. If possible, with practical application.

We have made amendments to the abstract and throughout the manuscript to focus the messaging and practical applications of the paper.

612 – More information is requested besides acknowledgments.

We are unsure what specific additional information is requested here. A financial disclosure statement and competing interests statement are provided on pages 1 and 2 of the submission file.

616 - Please revise the format of the manuscript considering the journal template and instructions for authors. Please check all details.

PLoS submission guidelines (https://journals.plos.org/plosone/s/submission-guidelines) have been closely reviewed and we are confident that the manuscript formatting adheres appropriately to these guidelines.

Please carefully revise English details throughout the manuscript.

Spelling, grammar and diction has been reviewed throughout the manuscript to ensure there are no errors and to clarify key messages of the paper.

Pages 43-49 – Please improve the figures quality. Please include the units in the axis.

Higher resolution versions of each figure are accessible through the hyperlinks at the top right corner of each figure page. All figures in the original submission meet the specified guidelines for quality in terms of dimensions and resolution (300 – 600 dpi). Although not always seen in self-organising maps, a label for the axis has now been included to clearly indicate that shading denotes a distance measure of arbitrary units between neighbouring points.

Responses to reviewer #2:

I thank the Authors and the Editors for the chance to review the present article. The manuscript investigates the influence of lightweight wearable resistance on whole-body coordination during sprints in Australian Rules football players. The manuscript is well structured and written, and I compliment the Authors for designing such a remarkable analysis. There are, however, some criticisms that should be addressed before the article can be deemed suitable for publication.

The introduction is informative and comprehensive about the topic. However, some paragraphs seem to be redundant, with the risk of making the section wordy and complicated to follow for the reader (e.g., lines 73-99). I would suggest streamlining the section and focusing more on the study rationale, which seems to be vague at this point. The purpose appears more in the direction of highlighting the possibility of the presented analysis framework being implemented in future larger studies rather than revealing possible effects in the presented dataset. It should be clarified whether the presented manuscr

---

## [Decision Letter · Decision Letter 1]

4 Jul 2024

PONE-D-23-30684R1The influence of lightweight wearable resistance on whole body coordination during sprint acceleration among Australian Rules football players.PLOS ONE

Dear Dr. Trounson,

Thank you for submitting your manuscript to PLOS ONE. After careful consideration, we feel that it has merit but does not fully meet PLOS ONE’s publication criteria as it currently stands. Therefore, we invite you to submit a revised version of the manuscript that addresses the points raised during the review process.

We look forward to receiving your revised manuscript.

Kind regards,

Laura-Anne Marie Furlong

Academic Editor

PLOS ONE

Reviewers' comments:

Reviewer's Responses to Questions

**Comments to the Author**

1. If the authors have adequately addressed your comments raised in a previous round of review and you feel that this manuscript is now acceptable for publication, you may indicate that here to bypass the “Comments to the Author” section, enter your conflict of interest statement in the “Confidential to Editor” section, and submit your "Accept" recommendation.

Reviewer #2: All comments have been addressed

Reviewer #3: (No Response)

2. Is the manuscript technically sound, and do the data support the conclusions?

Reviewer #2: Yes

Reviewer #3: Partly

3. Has the statistical analysis been performed appropriately and rigorously? 

Reviewer #2: Yes

Reviewer #3: No

4. Have the authors made all data underlying the findings in their manuscript fully available?

Reviewer #2: Yes

Reviewer #3: Yes

5. Is the manuscript presented in an intelligible fashion and written in standard English?

Reviewer #2: Yes

Reviewer #3: Yes

6. Review Comments to the Author

Reviewer #2: I thank the Authors for addressing all my comments, and I commend them again for the excellent work they have put into thoroughly designing and reviewing the manuscript.

Reviewer #3: Thank you for the opportunity to review “The influence of lightweight wearable resistance on whole body coordination during spring acceleration among Australian Rules football players”. I was not a part of the first review and my comments reflect the manuscript as currently constituted. The sample is limited, and it is difficult to generalize to the entire population of semi-pro Australian Rules football players, or to similar sports, yet it is analyzing a newer form of training. Also, it is using analyses that can be done for group or individual level comparisons. I do have some major and minor comments which need to be addressed, whether it be changes to the manuscript text and/or reviewer responses.

Major Comments

Introduction:

The introduction is good, overall. In lines 65-68, I recommend expanding your example to be biomechanically more accurate. I assume you mean the orientation of the 2D (or 3D) resultant GRF, which is influenced by the planes being measured, therefore, the AP GRF has largest effect on that orientation. If possible, spell it out more clearly.

You need to refer to adaptability and performance in the introduction.

Methods:

Study Design:

At least 8 weeks to perform all of the testing from familiarization through the WR conditions is a long time. Are there any fatigue effects as the season progresses? I see the control with being at least 48 hours post-match or team training but is there an accumulation from the season? Without WR as an influence, do players maintain the same performance with sprinting speed and acceleration as the season progresses?

Consider including details about the warmup that was provided or allowed prior to the maximum sprints in this section rather than Data collection.

I thought of this while reading the discussion, why wasn’t there a washout sprint. It would be interesting to see how the WR trials differ from sprint 1 without WR, and then in a washout (sprint 5), if some of the WR effects held.

Wearable resistance:

Did you measure the participants mass at each session? With such a large timespan, mass could change.

Did you determine a specific location on the thigh and shank to be consistent across sessions within participants, and the same relative location on a segment (e.g., middle third of shank) to be consistent across participants? I only ask because moment inertia involves more than the mass, distance is important to note as well.

Data collection:

Which dynamic mobility drills? Do you have a reference to indicate which drills and technique for them?

Who placed the markers on the participants, same person? Has that person been test for intra-session reliability on marker placements? If done by multiple people, has there been any inter-testing reliability performed?

Good work explaining the clustering analysis, SOM, and distance matrix. I think the explanations are clear.

Results:

The results are word-heavy because of the way it was analyzed and the small sample. Consider adding ranges to the data for Table 2 set to go along with means and SD, or range instead of SD because SD does not tell us as much with a small sample.

Discussion:

Lines 466-469: I can understand the desire to promote greater hip extension, but why would an athlete want transverse plane shoulder motion? I know they are examples based on the results, but maybe note that some motions are not wanted necessarily and overall, some motions being explored are not necessarily optimal.

Lines 476-483: How do you propose this individual level analysis? Also, to be ecologically driven, wouldn’t it be best to analyze on the field during practice and require some type of sensor, such as an IMU?

The information from Lines 503-508 needs to be woven into the introduction as well. It will help drive home the point that adaptability is good and leads to higher performance.

Consider making mention of adaptability in the context of injury prevention, not just performance.

In the limitations section, make mention of the study not being ecological valid because data was collected in a laboratory, not on the playing field in a game or practice environment.

Conclusions

Good section, clean advice at the end.

Notes on Figures

Beautiful figures overall. One note, please bold the axes units and titles for figure 7 (joint kinematic data).

Minor Comments

Introduction:

Line 74: “…often utilized deconstructed, part practice drills...” Something is missing, or the punctuation is in the wrong place. Or deconstructed what?

Line 79-80: “…by coaches to afford specific movement…”, consider changing “afford” to “influence”.

Line 81: “Both approaches” should be spelled out because I am not sure which two approaches you are referring to. I am guessing deconstructed and constraint, but I am certain.

Line 88: Remove “particular” because the segments are given after the comma.

Line 91: Remove “particular”

Line 92: “…are seen as being indicative of overload having occurred [].” Consider changing to “…indicate overload has occurred [].”

Methods:

Line 114: Include the “.0” for “72” to stay consistent with the other means.

Lines 118-120: How was that average distance and average running speed determined for these players?

Line 221: Add “cubic” in front of “spline” which I assume was the polynomial spline used.

Lines 228-229: Indicate if a researcher identified these points or if the V3D kinematic algorithm did.

Results:

Lines 361-362: “…, except for in the heavy anterior shank condition…” does not fit with the sentence and should be deleted.

Line 439: Include a comma between “condition pelvic”

Discussion:

Lines 506-515: Condense this section, it is superfluous, especially the two sentences from 511-515.

7. PLOS authors have the option to publish the peer review history of their article (what does this mean?). If published, this will include your full peer review and any attached files.

Reviewer #2: No

Reviewer #3: No

---

## [Author Response · Author response to Decision Letter 1]

20 Sep 2024

Dear Dr. Furlong,

We thank you again for the opportunity to re-revise our manuscript entitled “The influence of lightweight wearable resistance on whole body coordination during sprint acceleration among Australian Rules football players”. We would also like to thank Reviewer #2 for their support for the revised manuscript and Reviewer #3 for their insightful feedback and suggestions. Reviewer comments are included below with our responses provided following each comment. In the revised manuscript, adjustments to the methods and study design sections have been made to include further details of warm-up procedures and WR placement. Details of the benefits of adaptability for field-based athletes have been included in the introduction and discussion sections. Finally, additional limitations pertaining to ecological validity and potential training adaptations across the 8 week testing period have been included. We thank the reviewer for their considered feedback and feel the quality of the revised manuscript is substantially improved.

Best regards,

Karl Trounson

Responses to Reviewer #3:

Reviewer #3: Thank you for the opportunity to review “The influence of lightweight wearable resistance on whole body coordination during spring acceleration among Australian Rules football players”. I was not a part of the first review and my comments reflect the manuscript as currently constituted. The sample is limited, and it is difficult to generalize to the entire population of semi-pro Australian Rules football players, or to similar sports, yet it is analyzing a newer form of training. Also, it is using analyses that can be done for group or individual level comparisons. I do have some major and minor comments which need to be addressed, whether it be changes to the manuscript text and/or reviewer responses.

Major Comments

Introduction:

The introduction is good, overall. In lines 65-68, I recommend expanding your example to be biomechanically more accurate. I assume you mean the orientation of the 2D (or 3D) resultant GRF, which is influenced by the planes being measured, therefore, the AP GRF has largest effect on that orientation. If possible, spell it out more clearly.

We have amended the point highlighted to clarify that the anteroposterior component of GRFs resulting from foot strike is where meaningful differences between faster and slower athletes are observed. The revised version includes greater biomechanical detail to remove ambiguity associated with the original term ‘forward ground reaction forces’.

You need to refer to adaptability and performance in the introduction.

We have amended the introduction to make mention of adaptability and its utility in maintaining performance for field-based athletes under dynamic and varying constraints.

Methods:

Study Design: At least 8 weeks to perform all of the testing from familiarization through the WR conditions is a long time. Are there any fatigue effects as the season progresses? I see the control with being at least 48 hours post-match or team training but is there an accumulation from the season? Without WR as an influence, do players maintain the same performance with sprinting speed and acceleration as the season progresses?

It does not appear that there is a fatigue effect in relation to maximal speed performance across the course of several weeks in-season. In Australian Football, no clear trend is apparent in relation to distance covered above 90% of maximum velocity across an 8 week period [1]. Week-to-week changes appear to be match specific (presumably resulting from differences in weather conditions, tactics, ground dimensions, etc.) Among soccer players, however, repeated measures testing of 15 m [2], 35 m [3], and 50 m [4] sprints over the course of a competitive period consistently show fastest sprint times exhibited at the end of the season. This is likely due primarily to the frequency of exposure to sprint efforts in season, as well as possibly decreases in body fat across the course of a season. We have included an acknowledgement of this exposure effect as a potentially confounding factor in our limitations section.

Consider including details about the warmup that was provided or allowed prior to the maximum sprints in this section rather than Data collection.

Details of the warm-up have been moved to the study design section in line with the reviewer’s suggestion.

I thought of this while reading the discussion, why wasn’t there a washout sprint. It would be interesting to see how the WR trials differ from sprint 1 without WR, and then in a washout (sprint 5), if some of the WR effects held.

We wholly concur that this represents a worthwhile area of further exploration. Findings from weighted bat swinging in baseball suggest that temporal sequencing remains acutely affected when batters return to a normal weighted implement [5, 6]. It is likely that some movement after-effects manifest in acceleration and it would be of interest to know whether these are in the direction of, or opposed to, the coordination patterns exhibited with the WR. We believe, however, that this specific question warrants standalone research and would be too extensive to include in the present investigation.

Wearable resistance: Did you measure the participants mass at each session? With such a large timespan, mass could change.

Participant body mass was measured at the commencement of each session. Based on moment of inertia calculations and the smallest WR fraction (50 g), fluctuations of approximately 2 kg were needed for a 180 cm tall participant to warrant altering the loading magnitude in the heavy thigh conditions. In light thigh conditions and shank conditions larger magnitude body mass changes were required. In each of the relevant testing sessions, there were no fluctuations of sufficient magnitude to warrant alteration of the initial prescribed WR load.

Did you determine a specific location on the thigh and shank to be consistent across sessions within participants, and the same relative location on a segment (e.g., middle third of shank) to be consistent across participants? I only ask because moment inertia involves more than the mass, distance is important to note as well.

Yes, WR loads were situated with the middle of the fusiform shape placed at the midpoint of the relevant segment. We have sought to clarify this in the methods.

Data collection: Which dynamic mobility drills? Do you have a reference to indicate which drills and technique for them?

The dynamic mobility drills implemented have now been specified and a reference provided describing the technical execution of each drill.

Who placed the markers on the participants, same person? Has that person been test for intra-session reliability on marker placements? If done by multiple people, has there been any inter-testing reliability performed?

The same experienced biomechanics lab technician applied markers to participants throughout the study. While we did not undertake inter-session reliability testing, this data could be calculated among the participant group for some or all markers if deemed necessary. Given the capability of the technician to identify anatomical landmarks and the athletic physique of participants, consequential between-session variation in marker placement was/is not expected.

Good work explaining the clustering analysis, SOM, and distance matrix. I think the explanations are clear.

We thank the reviewer for their positive feedback on this section.

Results:

The results are word-heavy because of the way it was analyzed and the small sample. Consider adding ranges to the data for Table 2 set to go along with means and SD, or range instead of SD because SD does not tell us as much with a small sample.

Table 2 and S1 Table have been amended to report the data means and ranges in line with the reviewer’s suggestion.

Discussion:

Lines 466-469: I can understand the desire to promote greater hip extension, but why would an athlete want transverse plane shoulder motion? I know they are examples based on the results, but maybe note that some motions are not wanted necessarily and overall, some motions being explored are not necessarily optimal.

For early sprint acceleration, frontal and transverse plane motion have been identified as important contributors to shoulder, thoracic, and hip peak angular velocity [7]. Increases to the lever arm of the arm and leg through these coupled motions may facilitate the greater total impulse required during acceleration compared with maximum velocity sprinting. Therefore, we believe there is some merit in promoting transverse shoulder plane motion, however, we have added a note before the ‘limitations’ section that coaches should consider on a case basis whether accentuated movement in both the sagittal and transverse planes is desired.

Lines 476-483: How do you propose this individual level analysis? Also, to be ecologically driven, wouldn’t it be best to analyze on the field during practice and require some type of sensor, such as an IMU?

We have highlighted the specific analyses that were implemented to evaluate individual-level changes. However, we would like to clarify whether the reviewer is asking how we would propose individual level analyses be carried out by coaches in the field. We would also like to clarify the comment regarding ‘ecologically driven’ analysis. While we agree that for assessments to be ecologically valid they should be conducted on field during practice, we do not discount the value of performing assessments in less representative environments to facilitate adequate data collection.

The information from Lines 503-508 needs to be woven into the introduction as well. It will help drive home the point that adaptability is good and leads to higher performance.

A short section in the introduction has been added to convey that movement variability afforded by WR may be practically useful in training to develop adaptability, and that adaptability facilitates task execution in a wider array of scenarios. This is applicable for field-based athletes who must negotiate dynamic and changing constraints during match play.

Consider making mention of adaptability in the context of injury prevention, not just performance.

We have amended the section under the subheading ‘skill acquisition and coaching implications’ to include mention of the proposed functional role of adaptability in injury prevention.

In the limitations section, make mention of the study not being ecological valid because data was collected in a laboratory, not on the playing field in a game or practice environment.

As with the previous comment regarding ecological validity; while we make mention of skill acquisition approaches grounded in complex systems theory, we do not explicitly refer to ecological dynamics. We predominantly refer to constraints on the movement system, or constraints as movement control parameters, which we don’t necessarily feel have to be implemented in a representative environment if the goal is to increase action capabilities. Nevertheless, we have added a statement in the limitations section noting that because complex systems-based pedagogical approaches emphasise ecological validity in training, it is reasonable to propose that testing should occur in an ecologically valid setting as well.

Conclusions

Good section, clean advice at the end.

We thank the reviewer for their positive feedback on this section.

Notes on Figures

Beautiful figures overall. One note, please bold the axes units and titles for figure 7 (joint kinematic data).

Axes units and titles have been made bold for greater clarity in Figure 7.

Minor Comments

Introduction:

Line 74: “…often utilized deconstructed, part practice drills...” Something is missing, or the punctuation is in the wrong place. Or deconstructed what?

We have rephrased this to simply ‘skill deconstruction’, and hope this conveys more clearly the approach traditionally taken by strength and conditioning coaches in developing sprint technique.

Line 79-80: “…by coaches to afford specific movement…”, consider changing “afford” to “influence”.

Change has been made in line with reviewer suggestion.

Line 81: “Both approaches” should be spelled out because I am not sure which two approaches you are referring to. I am guessing deconstructed and constraint, but I am certain.

We have sought to clarify that the two approaches being referred to are (i) the use of constraints to shape movement in a particular direction in line with a desired movement outcome, and (ii) the use of constraints to induce variability generally and facilitate implicit exploration of movement patterns.

Line 88: Remove “particular” because the segments are given after the comma.

Change has been made in line with reviewer suggestion.

Line 91: Remove “particular”

Change has been made in line with reviewer suggestion.

Line 92: “…are seen as being indicative of overload having occurred [].” Consider changing to “…indicate overload has occurred [].”

Change has been made in line with reviewer suggestion.

Methods:

Line 114: Include the “.0” for “72” to stay consistent with the other means.

Change has been made in line with reviewer suggestion.

Lines 118-120: How was that average distance and average running speed determined for these players?

While no direct measurement of the study participants was performed, the range provided is based on available literature relating to semi-professional Australian Rules football players in-season. The relevant citations have been added here.

Line 221: Add “cubic” in front of “spline” which I assume was the polynomial spline used.

Change has been made in line with reviewer suggestion.

Lines 228-229: Indicate if a researcher identified these points or if the V3D kinematic algorithm did.

We have clarified that an event detection algorithm in Visual 3D set to identify the initial increase in toe marker vertical displacement was used to identify toe-off timepoints.

Results:

Lines 361-362: “…, except for in the heavy anterior shank condition…” does not fit with the sentence and should be deleted.

Change has been made in line with reviewer suggestion.

Line 439: Include a comma between “condition pelvic”

Comma is now included as per reviewer’s suggestion.

Discussion:

Lines 506-515: Condense this section, it is superfluous, especially the two sentences from 511-515.

The authors agree with this feedback and this section has been condensed accordingly.

1. Freeman BW, Talpey SW, James LP, Rayner RJ, Young WB. Preseason and in-season high-speed running demands of 2 professional Australian Rules football teams. Sports Health. 2024:19417381241265114. Epub 20240822. doi: 10.1177/19417381241265114. PubMed PMID: 39171493.

2. Caldwell BP, Peters DM. Seasonal variation in physiological fitness of a semiprofessional soccer team. J Strength Cond Res. 2009;23(5):1370-7. doi: 10.1519/JSC.0b013e3181a4e82f. PubMed PMID: 19620929.

3. Evangelos B, Lefteris M, Aristotelis G, Ioannis G, Natalia K. Aerobic and anaerobic capacity of professional soccer players in annual macrocycle. J Phys Educ Sport. 2016;16(2):527.

4. Ostojic SM. Seasonal alterations in body composition and sprint performance of elite soccer players. J Exerc Physiol. 2003;6(3):11-4.

5. Nakamoto H, Ishii Y, Ikudome S, Ohta Y. Kinesthetic aftereffects induced by a weighted tool on movement correction in baseball batting. Hum Mov Sci. 2012;31(6):1529-40. Epub 2012/06/16. doi: 10.1016/j.humov.2012.04.005. PubMed PMID: 22698835.

6. Ohta Y, Ishii Y, Ikudome S, Nakamoto H. Warm-up with weighted bat and adjustment of upper limb muscle activity in bat swinging under movement correction conditions. Percept Mot Skills. 2014;118(1):96-113. doi: 10.2466/23.25.PMS.118k15w8. PubMed PMID: 24724516.

7. Slawinski J, Bonnefoy A, Ontanon G, Lévêque J-M, Miller C, Riquet A, et al. Segment-interaction in sprint start: Analysis of 3D angular velocity and kinetic energy in elite sprinters. J Biomech. 2010;43(8):1494-502.

---

## [Decision Letter · Decision Letter 2]

22 Oct 2024

The influence of lightweight wearable resistance on whole body coordination during sprint acceleration among Australian Rules football players.

PONE-D-23-30684R2

Dear Dr. Trounson,

We’re pleased to inform you that your manuscript has been judged scientifically suitable for publication and will be formally accepted for publication once it meets all outstanding technical requirements.

Kind regards,

Laura-Anne Marie Furlong

Academic Editor

PLOS ONE

Additional Editor Comments (optional):

Reviewers' comments:

Reviewer's Responses to Questions

**Comments to the Author**

1. If the authors have adequately addressed your comments raised in a previous round of review and you feel that this manuscript is now acceptable for publication, you may indicate that here to bypass the “Comments to the Author” section, enter your conflict of interest statement in the “Confidential to Editor” section, and submit your "Accept" recommendation.

Reviewer #2: All comments have been addressed

Reviewer #3: All comments have been addressed

2. Is the manuscript technically sound, and do the data support the conclusions?

Reviewer #2: Yes

Reviewer #3: Yes

3. Has the statistical analysis been performed appropriately and rigorously? 

Reviewer #2: Yes

Reviewer #3: Yes

4. Have the authors made all data underlying the findings in their manuscript fully available?

Reviewer #2: Yes

Reviewer #3: Yes

5. Is the manuscript presented in an intelligible fashion and written in standard English?

Reviewer #2: Yes

Reviewer #3: Yes

6. Review Comments to the Author

Reviewer #2: I thank the Authors for addressing all my comments, and I commend them again for the excellent work they have put into thoroughly designing and reviewing the manuscript.

Reviewer #3: Thank you for addressing all of my comments. I think it is a strong manuscript and is deserving of publication.

7. PLOS authors have the option to publish the peer review history of their article (what does this mean?). If published, this will include your full peer review and any attached files.

Reviewer #2: No

Reviewer #3: No

---

## [Editor Report · Acceptance letter]

28 Oct 2024

PONE-D-23-30684R2 

PLOS ONE

Dear Dr. Trounson, 

I'm pleased to inform you that your manuscript has been deemed suitable for publication in PLOS ONE. Congratulations! Your manuscript is now being handed over to our production team.

Kind regards, 

on behalf of

Dr. Laura-Anne Marie Furlong 

Academic Editor

PLOS ONE